# Thai psychiatrists and burnout: A national survey

**Neshda Nimmawitt**[ID][☯], **Kamonporn Wannarit**[☯], **Pornjira Pariwatcharakul**[ID]*

Department of Psychiatry, Faculty of Medicine Siriraj Hospital, Mahidol University, Bangkok, Thailand

☯ These authors contributed equally to this work.
* pornjira.par@mahidol.edu

## Abstract

### Objectives

To explore the prevalence and factors that contribute to burnout among Thai psychiatrists.

### Background

The practice of psychiatry can lead to emotional fatigue. As rates of emotional illness in Thailand continue to climb, increasing demands are placed on a limited number of psychiatrists. This can lead to burnout, and multiple negative physical and mental health outcomes.

### Materials and methods

Electronic questionnaires were sent to all 882 Thai psychiatrists and residents via a private social media group managed by the Psychiatric Association of Thailand. The questionnaire included demographic data, the Maslach Burnout Inventory (MBI), the Proactive Coping Inventory, and questions about strategies that Thai psychiatrists believed reduce/prevent burnout.

### Results

Questionnaires were sent and 227 (25.7%) responded. According to MBI, 112 (49.3%) of respondents reported high level of emotional exhaustion, and 60 (26.4%) had a high level of depersonalization. Nearly all respondents (99.6%) maintained a high level of personal accomplishment. Working more than 50 hours per week (p = 0.003) and more patients per day (p = 0.20) were associated with higher levels of burnout. Feeling satisfied with work (p<0.001) and having a good support system from family (p = 0.027) and colleagues (p = 0.033) were associated with lower levels of burnout. The coping mechanisms related to lower levels of burnout included more emotional support seeking (p = 0.005), more proactive coping (p = 0.047), and less avoidance (p = 0.005).

### Conclusions

Compared to a previous study on burnout among Thai psychiatrists in 2011, in this study, the prevalence of high levels of burnout had increased dramatically from 17.1% to 49.3%.

**Data Availability Statement:** All relevant data are within the manuscript and its Supporting Information files.

**Funding:** NN, KP and PP received a research grant from the Psychiatric Association of Thailand

(http://psychiatry.or.th). No grant number was issued. The funders had no role in study design, data collection and analysis, decision to publish, or preparation of the manuscript.

**Competing interests:** The authors have declared that no competing interests exist.

An intervention to decrease workload, strengthen social support and encourage proactive coping mechanisms may be beneficial for relieving burnout.

## Introduction

Burnout is a syndrome of emotional exhaustion (feelings of mental fatigue and a lack of positive energy), depersonalization (negative attitudes toward work and patients), and a negative view of personal accomplishment (dissatisfied feelings about self-efficacy) [1]. Burnout may have physical, mental and occupational consequences. Physical effects of burnout include metabolic syndromes, chronic somatic symptoms, severe injuries and mortality under the age of 45 years [2]. Mental health outcomes include insomnia, depressive symptoms, the use of psychotropic medications and even hospitalization [2, 3]. Burnout can result in job dissatisfaction and career deterioration [2].

Psychiatrists have daily interactions with mentally ill patients and are at high risk of burnout [3]. A 2011 national survey of Thai psychiatrists demonstrated that nearly half (44.7%) of the respondents had moderate to high levels of burnout [4], which was consistent with findings from other countries. Multiple studies have reported high levels of emotional exhaustion in psychiatrists from New Zealand, Finland, Canada, the United Kingdom and the United States [5–9], whereas Psychiatrists from Japan, Austria, Italy and Germany experience low to moderate levels of emotional exhaustion [9, 10].

Factors associated with burnout among psychiatrists include the lack of support systems, the negative characteristics of patients and their relatives, suicidal death in their patients, long working hours, limited experience in the field, and excessive workload [11]. In contrast, factors associated with job satisfaction include work autonomy, supportive environments, and being valued [11].

Coping mechanisms are the strategies that people use to manage their stress [12]. While traditional, reactive conceptions of coping mechanisms focus on dealing with preexisting stress, proactive coping mechanisms emphasize anticipating potential stress and planning to prevent undesirable outcomes of the situation [13]. Proactive coping mechanisms are associated with a low level of burnout [14]. Therefore, we aimed to explore burnout, factors associated with burnout and the relationship between coping style and burnout in Thai psychiatrists.

## Materials and methods

### Respondents and procedure

In June 2018 online questionnaires were sent to all 882 Thai psychiatrists and psychiatric residents via a closed social media group operated by the Psychiatric Association of Thailand. Participation in the study was voluntary, anonymous and no financial or other incentives were provided. The Siriraj Institutional Review Board approved the study protocol (Certificate of Approval Number: Si 333/2018).

### Questionnaire

The questionnaire comprised four domains: (1) sociodemographic data and job satisfaction, (2) burnout, (3) coping mechanisms, and (4) strategies that Thai psychiatrists believed could help reduce their burnout. (S1 and S2 Appendices).

**Sociodemographic data and work-related aspects.** The participants were requested to rate their job satisfaction and income satisfaction on a score from 0 to 10 (0 = unsatisfied,

10 = most satisfied). Furthermore, the participants were asked to categorize the quality of their perceived social supports (e.g. family, friends) as good, poor, or unavailable.

**Burnout.**    Burnout was measured using the Thai version of the Maslach Burnout Inventory (MBI) questionnaire [1, 15]. The MBI consists of 22 items divided into three dimensions: emotional exhaustion (feelings of being emotionally overextended and exhausted by one's work), depersonalization (unsympathetic and impersonal responses toward the recipients of one's care or service), and personal accomplishment (feelings of competence and successful achievement in one's work with people) [1]. For the emotional exhaustion and depersonalization subscales, higher mean scores correspond to higher degrees of burnout (emotional exhaustion score: 0–16 = low, 17–26 = moderate, > 26 = high; depersonalization score: 0–6 = low, 7–12 = moderate, > 12 = high). Lower mean scores of personal accomplishment correspond to higher degrees of burnout (personal accomplishment score: > 38 = low, 32–38 = moderate, 0–31 = high). The Cronbach's alpha coefficient of each domain in the Thai version of MBI is between 0.65–0.92 [15].

**Coping mechanisms.**    The Thai version of the Proactive Coping Inventory (PCI) questionnaire [16, 17] is a multidimensional instrument that consists of 55 items divided into seven scales. The literature supports that burnout is associated with two major coping strategies: problem-focused coping and emotional-focused coping. Problem-focused coping includes proactive coping, directive coping, and plan resolution strategies. Emotional-focused coping covers positive reappraisal, seeking social support, and avoidance [18–20]. Because each PCI scale is independent from each other, the authors selected four scales to make the questionnaire more concise. Scale #1 is proactive coping defined as the combination of autonomous goal setting with self-regulatory goal attainment cognition and behavior. Scale #2 is strategic planning defined as the process of generating a goal-oriented schedule of action in which extensive tasks are broken down into more manageable tasks. Scale #3 is emotional support seeking defined as the regulation of temporary emotional distress by disclosing feelings to others, evoking empathy and seeking companionship from one's social network. Last, Scale #4 is avoidance coping defined as the illusion of action in a demanding situation by delaying action. There was no cut-off point for each scale; a higher score indicated a greater degree of each coping mechanism. The Cronbach's alpha coefficient of each PCI domain in the Thai version is between 0.70–0.81 [17].

**Strategies that Thai psychiatrists believed could ameliorate their burnout.**    Guided by a review of the literature, the investigators identified 21 issues that psychiatrists are concerned about. (Part 4 of S2 Appendix). Each issue was rated from 1 to 10: 1 was "should be reduced as much as possible", 5 was "no need to change", and 10 was "should be increased as much as possible". Space to write additional comments was provided at the end of the questionnaire.

## Statistical analysis

Data were analyzed using descriptive statistics, t-tests, chi-square tests and ANOVAs. Spearman correlations were calculated between the MBI subscales and the PCI subscales. We performed multiple linear regressions to test the correlations between individual sociodemographic factors and MBI subscales. The missing values were treated as missing. All statistical analyses were conducted using IBM SPSS Statistics version 21 (IBM Corporation, Armonk, NY, USA). The two-sided level of significance was set at $p < 0.05$.

The associations between individual sociodemographic factors and MBI scores were determined by using multiple linear regressions. The factors entered into the model included sociodemographic factors and coping mechanisms according to the PCI subscales. Variables were included by a stepwise selection procedure (entry $p < 0.05$).

# Results

## Sociodemographic and work-related aspects

The response rate was 25.7% (n = 227). The majority of participants were female (70.5%) and were aged between 25 and 64 years (mean = 36.4; SD = 8.4) with an average of 8.9 years of experience as a psychiatrist. Most of the respondents were general psychiatrists (58.1%), single (55.9%) and had no children (71.9%). (Table 1).

## Job satisfaction and quality of social support

The mean job satisfaction score was rated as 6 out of 10 (SD 2.49), and the mean income satisfaction score was rated as 6.81 out of 10 (SD 2.04). When facing stress, the respondents chose to consult their friends, themselves, their partners, their family members, the chiefs of the psychiatric department, and social media. Good family support was available to 93.8% of the respondents, while 85% of respondents stated that they received good support from psychiatrist friends and other colleagues. Almost two third (64%) of respondents reported receiving good support from their department chiefs and doctors in other departments. In contrast, 46.7% of respondents reported they received good support from the hospital administrative staff.

**Burnout.** According to the MBI, 112 (49.3%) of the respondents had a high level of emotional exhaustion, whereas 60 (26.4%) had a high level of depersonalization. The mean MBI score for emotional exhaustion was 26.26 (SD 12.521), and the mean score for depersonalization was 7.96 (SD 7.137). However, most of the respondents (99.6%) still had a high level of personal accomplishment. The mean score for personal accomplishment was 9.7 (SD 6.469). (Fig 1).

**Correlation between coping mechanisms and MBI subscales.** There were statistically significant correlations between each coping mechanism and burnout. (Table 2). Higher skill levels of proactive coping, strategic planning and emotional seeking were associated with less emotional exhaustion, less depersonalization, and greater personal accomplishment (r = -0.168 to -0.433). On the other hand, higher avoidance coping scores were associated with more emotional exhaustion (r = 0.241), more depersonalization (r = 0.300), and less personal accomplishment (r = 0.189) (Table 2).

## Factors associated with burnout

Emotional exhaustion was increased in participants who worked more than 50 hours per week (B = 2.958, p = 0.003). (Table 3) A higher caseload was associated with higher levels of emotional exhaustion (B = 0.07, p = 0.02). However, greater work satisfaction (B = -3.035, p<0.001) and more support from families (B = -3.263, p = 0.041) were associated with less emotional exhaustion. Emotional support seeking (B = -0.569, p = 0.008) was the only coping mechanism associated with less emotional exhaustion.

Depersonalization was higher in the participants who had a higher number of patients per day (B = 0.092, p = 0.000) and who used more avoidance coping mechanisms (B = 0.634, p = 0.005). In contrast, having good support from colleagues (B = -1.468, p = 0.033), being satisfied with work (B = -1.002, p<0.001), and having more than one child (B = -1.301, p = 0.018) were associated with less depersonalization. Additionally, the proactive coping mechanism (B = -0.150, p = 0.047) was found more often in individuals with less depersonalization. Finally, being satisfied with work (B = -1.092, p<0.001) and proactive coping (B = -0.275, p<0.001) were associated with more personal accomplishment.

**Table 1. Sociodemographic variables and work-related aspects of participants.**

| | Participants (n = 227) (%) [a] | All Thai psychiatrists [b] (n = 882) (%) |
|---|---|---|
| Age (years), mean ± SD | 36.4 ± 8.4 | N/A |
| Sex | | |
| Male | 67 (29.5) | 395 (44.8) |
| Female | 160 (70.5) | 487 (55.2) |
| Marital status | | |
| Single | 124 (55.9) | N/A |
| Married | 89 (40.1) | N/A |
| Widow/divorced | 9 (4.1) | N/A |
| Number of children | | |
| None | 161 (71.9) | N/A |
| 1 | 27 (12.1) | N/A |
| >1 | 36 (16.1) | N/A |
| Experience as a psychiatrist (years), mean ± SD | 8.9 ± 9.1 | 17 |
| Position | | |
| General psychiatrist | 132 (58.1) | 614 (69.6) |
| Child and adolescent psychiatrist | 39 (17.2) | 158 (17.9) |
| Psychiatry resident | 56 (24.7) | 110 (12.5) |
| Death of patients who committed suicide | | |
| None | 124 (54.6) | N/A |
| ≤ 1 month | 6 (2.6) | N/A |
| > 1 month—< 1 year | 31 (13.7) | N/A |
| ≥ 1 year | 66 (29.1) | N/A |
| Working hours per week | | |
| < 40 hours | 38 (16.8) | N/A |
| 40 - < 50 hours | 112 (49.6) | N/A |
| ≥ 50 hours or more | 76 (33.6) | N/A |
| Number of shifts per month, mean ± SD | 8.5 ± 5.7 | N/A |
| Number of patients per day, mean ± SD | 27.1 ± 21.1 | N/A |
| Days off per month | | |
| 0–2 | 23 (10.1) | N/A |
| 3–5 | 76 (33.5) | N/A |
| 6–8 | 83 (36.6) | N/A |
| 9–10 | 41 (18.1) | N/A |
| >10 | 4 (1.8) | N/A |
| Workplace | | |
| Regional hospital | 36 (15.9) | 127 (14.4) |
| General hospital | 48 (21.1) | 158 (17.9) |
| Community hospital | 7 (3.1) | 4 (0.5) |
| Psychiatric hospital | 45 (19.8) | 151 (17.1) |
| Medical school/Teaching hospital | 85 (37.4) | 195 (22.1) |
| Private hospital/clinic | 60 (26.4) | 99 (11.2) |
| Unknown | 0 (0) | 148 (16.8) |

[a]. Percentage of the total number of valid values for each variable.
[b]. Data from the registration office of the Psychiatric Association of Thailand

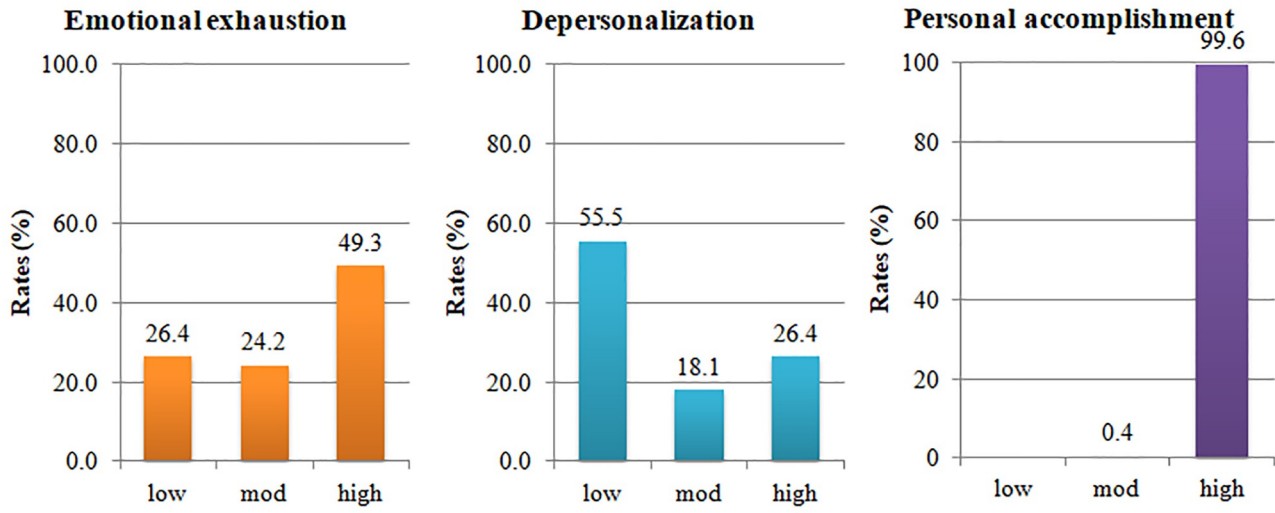

**Fig 1. Burnout rates according to the Maslach Burnout Inventory (MBI) subscales.**

**Strategies that could help reduce burnout.** Respondents reported they would like to increase workplace welfare, number of staff members, support from the head of the department, the departmental budget (median = 8; range 1–10), followed by good relationship among team members (median = 7; range 2–10), good relationship with the head of the department, support for new projects and innovation in the department, participation in changing of the organization, the income, workplace equipment and training for psychiatrists and team colleagues (median = 7; range 1–10), good relationship among psychiatrists (median = 6.5; range 1–10) and number of days off (median = 6; range 1–10). Respondents indicated they would like to decrease the amount of paperwork (median = 3; range 1–10) and the number of patients per day (median = 4; range 1–10). Other strategies are to change the number of working hours per day, number of shifts, administrative role in the department and the hospital as well as the amount of general practice work, e.g., emergency room shift, general patient examination (median = 5; range 1–10).

## Discussion

Psychiatrists have a high risk of burnout due to their work. A 2011 study from Thailand [4], reported that 17.1% of psychiatrists had high-level emotional exhaustion compared to 49.3% in this survey. Additionally, the mean emotional exhaustion score increased from 16.4 to

**Table 2. Correlation between coping mechanisms and Maslach Burnout Inventory (MBI) subscales.**

| Maslach Burnout Inventory (MBI) subscales | Coping mechanisms (r) | | | |
|---|---|---|---|---|
| | Proactive coping | Strategic planning | Emotional support seeking | Avoidance coping |
| Emotional exhaustion | -.370** | -.206** | -.298** | .241** |
| Depersonalization | -.385** | -.185** | -.168* | .300** |
| Personal accomplishment | -.433** | -.250** | -.197** | .189** |

r, Spearman correlation coefficient;

* Correlation is significant at the 0.05 level (2-tailed);

** Correlation is significant at the 0.01 level (2-tailed)

**Table 3. Factors associated with burnout: Multiple linear regression analysis.**

| Emotional exhaustion | Unstandardized coefficients | | 95% CI | p |
|---|---|---|---|---|
| | B | Std. Error | | |
| Working more than 50 hours per week | 2.803 | 0.946 | 0.937; 4.668 | 0.003 |
| Number of patients per day | 0.070 | 0.030 | 0.011; 0.129 | 0.020 |
| Good support from families | -3.432 | 1.541 | -6.469; -0.394 | 0.027 |
| Being satisfied with work | -3.323 | 0.333 | -3.978; -2.667 | <0.001 |
| Emotional support seeking | -0.571 | 0.202 | -0.969; -0.174 | 0.005 |
| Depersonalization | | | | |
| Avoidance coping mechanism | 0.634 | 0.225 | 0.191; 1.077 | 0.005 |
| Number of patients per day | 0.092 | 0.092 | 0.055; 0.130 | <0.001 |
| Good support from colleagues | -1.468 | 0.686 | -2.820; -0.116 | 0.033 |
| Having more than one child | -1.301 | 0.544 | -2.374; -0.228 | 0.018 |
| Being satisfied with work | -1.002 | 0.221 | -1.438; -0.566 | <0.001 |
| Proactive coping mechanism | -0.150 | 0.075 | -0.298; -0.002 | 0.047 |
| Personal accomplishment | | | | |
| Being satisfied with work | -1.092 | 0.210 | -1.505; -0.679 | <0.001 |
| Proactive coping mechanism | -0.275 | 0.069 | -0.412; -0.139 | <0.001 |

Variables included by a stepwise selection procedure (entry p < 0.05).

26.26. High-level depersonalization also increased from 5.5% in 2011 [4] to 26.4%, and the mean depersonalization score increased from 3.45 to 7.96. Our study showed a considerably higher burnout rate compared to other studies (19%-40%) among psychiatrists using the MBI questionnaire [21, 22].

Higher levels of emotional exhaustion was associated with working for more than 50 hours/week and treating more patients per day. In a study by Kumar in 2010, working long hours had the most significant association with a high emotional exhaustion score [23]. A study in psychiatric trainees from 22 countries arrived a similar conclusion [24]. Caring for higher numbers of patients per day was also found to be associated with higher emotional exhaustion and depersonalization. A 2018 systematic review and meta-analysis indicated that higher caseloads were associated with burnout in many studies [21]. In our study, higher caseload and working long hours were also related to higher levels of emotional exhaustion, but more cases were not associated with more working hours (F = 3.020, p = 0.051). It is possible that having more patients can cause psychiatrists to work longer hours. However, seeing fewer patients with challenging problems may need longer working hours as well. Further details are needed to explain this relationship.

Being satisfied with work, receiving support from families, and practicing emotional support seeking were associated with lower levels of emotional exhaustion. Being satisfied with work seemed to lower the level of burnout, both in terms of emotional exhaustion and depersonalization. This finding was consistent with previous studies in Thailand [4], Switzerland [9], New Zealand [25], the United Kingdom [26], Sweden [27], and the United States [28]. Feeling valued, having a variety of tasks and being supported in the clinical oversight role contribute to job satisfaction [28].

More support from families and good support from colleagues were also significant factors. A previous national survey in Thailand found that a lack of support from colleagues was associated with higher emotional exhaustion scores [4]. Furthermore, a study in Japanese psychiatrists showed that social support was the strongest factor associated with low burnout rates

[10]. Our result was consistent with a study in Taiwan that showed that greater workplace social support was associated with lower depression scores in physicians [29]. In addition, a survey among Finnish psychiatrists reported that an inability to consult a colleague and a lack of work supervision increased burnout [6]. A 1999 meta-analysis reported on the role of social support in the process of work stress. Social support reduced workplace strain, attenuated perceived stressors, and moderated the stressor-strain relationship [30].

Our findings support the value of emotional support seeking and receiving social support to reduce burnout. This is consistent with a study by Looney et al in 1980, which found that getting support from others was an effective coping mechanism for stress management [31]. Dallender et al noted that the most commonly used coping mechanism among psychiatrists in the United Kingdom was seeking support from loved ones and colleagues [32]. In our study, using more proactive coping mechanisms and less avoidance coping mechanisms was associated with lower burnout levels. This finding suggests that these coping mechanisms might be helpful in managing the stress levels of psychiatrists and psychiatry residents.

Despite high emotional exhaustion and depersonalization scores, most participants still had a high sense of personal accomplishment. This was also seen in a 2018 systematic review and meta-analysis of burnout among mental health professionals that reported overall mean score of 21.11 for emotional exhaustion, 6.76 for depersonalization and 34.60 for personal accomplishment [21]. In Maslach's study in 1987, the Personal Accomplishment subscale was independent of the other subscales [1]. In our study, being satisfied with work and a proactive coping mechanism predicted more personal accomplishment.

Participants' perspectives towards the strategies that could help reduce burnout are consistent with results in other studies [3–10]. However, the wide range of scores suggests that each psychiatrist needs different strategies to reduce burnout. This is in line with the need of other healthcare professionals [33]. A review on preventing occupational stress in healthcare workers suggests that interventions should focus on stressors that are specific to each organization [33].

One limitation of our study was the small number of participants, which may limit the generalizability of our findings. We attempted to compare the characteristics of the respondents with the non-responders, but there are no official data of the general characteristics of Thai psychiatrists in the whole country. We did obtain data from the registration office of the Psychiatric Association of Thailand. (Table 1). The Psychiatric Association data suggest that 45% of Thai psychiatrists while 30% of our respondents were male. The Psychiatric Association reports that the mean years of experience of all psychiatrists is 17 years, while our respondents reported a mean of 8.9 years' experience. Moreover, since there are only 882 psychiatrists in Thailand, 112 (49.3%) of the respondents with a high level of burnout accounted for approximately 12.7% of all psychiatrists in the country.

Our survey suggests that high levels of burnout are present and may be increasing in Thai psychiatrists. There is an urgent need to address this situation. We did not directly investigate the details of longer working hours and suggest exploring this issue in future studies. Our findings have been shared with the Psychiatric Association of Thailand, a professional body that helps establish work policies for psychiatrists in Thailand. To our knowledge, this is the first study to examine the correlation between coping mechanisms and burnout in Thai psychiatrists.

## Conclusion

Levels of burnout among Thai psychiatrists are high and may be increasing. Proactive coping strategies including emotional support seeking should be taught during residency training and

encouraged in the workplace. Efforts to limit working hours to less than 50 hours per week and to improve social supports are needed. The outcomes of burnout, such as the impact on work efficacy and physical health problems merit further study.

## Supporting information

**S1 Appendix. Thai version Questionnaire.**
(DOCX)

**S2 Appendix. English version Questionnaire.**
(DOCX)

**S1 Dataset. Thai version minimal dataset.**
(XLSX)

**S2 Dataset. English version minimal dataset.**
(XLSX)

## Acknowledgments

We would like to thank all respondents for kindly dedicating their time to completing our surveys. We appreciate Narathip Sanguanpanich for her advice regarding the statistical analysis. Additionally, we are grateful to Mark Simmerman, PhD and Siraphat Taesuwan, PhD, for their comments on and proofreading of this manuscript. This study was funded by the Psychiatric Association of Thailand. The funding source had no role in the design or implementation of the study. The authors (KW and PP) were supported by the Chalermphrakiat Grant, Faculty of Medicine Siriraj Hospital, Mahidol University. Some parts of this study were presented in a poster presentation at the 19th World Psychiatric Association World Congress of Psychiatry 2019.

## Author Contributions

**Conceptualization:** Neshda Nimmawitt, Kamonporn Wannarit, Pornjira Pariwatcharakul.

**Data curation:** Neshda Nimmawitt.

**Funding acquisition:** Pornjira Pariwatcharakul.

**Methodology:** Kamonporn Wannarit, Pornjira Pariwatcharakul.

**Project administration:** Neshda Nimmawitt.

**Resources:** Pornjira Pariwatcharakul.

**Supervision:** Kamonporn Wannarit, Pornjira Pariwatcharakul.

**Validation:** Kamonporn Wannarit.

**Visualization:** Neshda Nimmawitt.

**Writing – original draft:** Neshda Nimmawitt.

**Writing – review & editing:** Kamonporn Wannarit, Pornjira Pariwatcharakul.

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
