## [Decision Letter · Decision Letter 0]

18 Dec 2019

PONE-D-19-31990

Thai psychiatrists and burnout: a national survey

PLOS ONE

Dear Dr. Pariwatcharakul,

Thank you for submitting your manuscript to PLOS ONE. After careful consideration, we feel that it has merit but does not fully meet PLOS ONE’s publication criteria as it currently stands. Therefore, we invite you to submit a revised version of the manuscript that addresses the points raised during the review process.

Your paper has been assessed by one acknowledged expert in the field, who remarked the potential of the study, but asks for some improvements related to, e.g., additional analyses on the information used for contrasting the results, the scale selection and other further rationales that are required from you. Moreover, other comments -most of them relatively minor- should be also addressed and responded in the rebuttal letter.

Please see below. the full set of comments provided by the reviewer.

We would appreciate receiving your revised manuscript by Feb 01 2020 11:59PM. To enhance the reproducibility of your results, we recommend that if applicable you deposit your laboratory protocols in protocols.io, where a protocol can be assigned its own identifier (DOI) such that it can be cited independently in the future. For instructions see: http://journals.plos.org/plosone/s/submission-guidelines#loc-laboratory-protocols

We look forward to receiving your revised manuscript.

Kind regards,

Sergio A. Useche, Ph.D.

Academic Editor

PLOS ONE

Journal Requirements:

2. Please include additional information regarding the survey or questionnaire used in the study and ensure that you have provided sufficient details that others could replicate the analyses. For instance, if you developed a questionnaire as part of this study and it is not under a copyright more restrictive than CC-BY, please include a copy, in both the original language and English, as Supporting Information. In particular, the 20 issues included in the fourth section of the questionnaire “Strategies which Thai psychiatrists believed could ameliorate their burnout” should be specified.

Reviewers' comments:

Reviewer's Responses to Questions

**Comments to the Author**

1. Is the manuscript technically sound, and do the data support the conclusions?

Reviewer #1: Yes

2. Has the statistical analysis been performed appropriately and rigorously? 

Reviewer #1: Yes

3. Have the authors made all data underlying the findings in their manuscript fully available?

Reviewer #1: No

4. Is the manuscript presented in an intelligible fashion and written in standard English?

Reviewer #1: Yes

5. Review Comments to the Author

Reviewer #1: This is a second national burnout survey of Thai psychiatrists. The first national survey published in a Thai journal in 2011 by Lerthattasilp et al received a similarly low response rate of 27.8% (181/650), compared with the present study that has 27.15% (227/836). Despite the low response rate, the findings are still useful but characteristics of the responders should be compared with that of the non-responders if possible. Alternatively, the characteristics of all Thai psychiatrists is avaiable in the national registry and should be presented alongside with the characteristics of the responders to this survey.

While the first national survey used only the Maslach Burnout Inventory (MBI), the present survey added the Proactive Coping Inventory (PCI) in order to investigate the relationship between coping style and burnout in Thai psychiatrists. The authors need to provide a rationale how these 4 of the 7 PCI scales were selected: proactive coping, strategic planning, emotional support seeking, and avoidance coping.

The correlation between MBI and PCI subscales were assessed by using Pearson's correlation coefficients. As the MBI contains 22 items in 3 subscales whereas the PCI contains 55 items in 7 subscales (but 26 items in 4 subscales were selected in this study, normality of both variables had to be checked.

6. PLOS authors have the option to publish the peer review history of their article (what does this mean?). If published, this will include your full peer review and any attached files.

Reviewer #1: Yes: Assist. Prof. Dr. Krit Pongpirul, MD, MPH, PhD.

---

## [Author Response · Author response to Decision Letter 0]

31 Jan 2020

Dear Assist. Prof. Dr. Krit Pongpirul,

We are very thankful to you for the helpful comments and advice. We have addressed all comments as requested. Attached please kindly find a copy of the revised manuscript entitled “Thai psychiatrists and burnout: a national survey” (Ms. No. PONE-D-19-31990) by Neshda Nimmawitt, Kamonporn Wannarit, and Pornjira Pariwatcharakul, which we are submitting for publication as a Research Article in the PLOS ONE. 

We are explaining all the changes made and responding in detail to each point, as follows.

1) This is a second national burnout survey of Thai psychiatrists. The first national survey published in a Thai journal in 2011 by Lerthattasilp et al received a similarly low response rate of 27.8% (181/650), compared with the present study that has 27.15% (227/836). Despite the low response rate, the findings are still useful but characteristics of the responders should be compared with that of the non-responders if possible. Alternatively, the characteristics of all Thai psychiatrists is available in the national registry and should be presented alongside with the characteristics of the responders to this survey.

We really appreciated your suggestion. Accordingly, we tried to find the characteristics of all Thai psychiatrists, but there were no official data available. We requested raw data from the registration office of the Psychiatric association of Thailand and categorized into the demographic data in Table 1, then compared with the characteristics of the respondents. As the provided data were limited and outdated, we clarified the reliability of the data in our manuscript. We added the following sentences to the 7th paragraph of the Discussion section. 

 “One limitation of our study was the small number of participants, which may limit the generalizability of our findings. We attempted to compare the characteristics of the respondents with the non-responders, but there are no official data of the general characteristics of Thai psychiatrists in the whole country. We did obtain raw data from the registration office of the Psychiatric Association of Thailand. (Table 1) However, these data were outdated, incomplete and may not be reliable. The Psychiatric Association data suggest that 45% of Thai psychiatrists while 30% of our respondents were male. The Psychiatric Association reports that the mean years of experience of all psychiatrists is 18 years, while our respondents reported a mean of nine years experience.” 

2) While the first national survey used only the Maslach Burnout Inventory (MBI), the present survey added the Proactive Coping Inventory (PCI) in order to investigate the relationship between coping style and burnout in Thai psychiatrists. The authors need to provide a rationale how these 4 of the 7 PCI scales were selected: proactive coping, strategic planning, emotional support seeking, and avoidance coping.

Thank you for this comment. We modified the Materials and Methods section to emphasize the rationale how we selected these scales. We added the following sentences to the 5th paragraph of the Materials and Methods section.

 “The literature supports that burnout is associated with two major coping strategies. Problem-focused coping includes proactive coping, directive coping, and plan resolution strategies. Emotional-focused coping covers positive reappraisal, seeking social support, and avoidance [18-20]. Therefore, we selected only 4 scales to make the questionnaire more concise.”

3) The correlation between MBI and PCI subscales were assessed by using Pearson's correlation coefficients. As the MBI contains 22 items in 3 subscales whereas the PCI contains 55 items in 7 subscales (but 26 items in 4 subscales were selected in this study, normality of both variables had to be checked.

Thank you for pointing this out. We agree with this comment. We checked both of the variables and the results showed that both of them were non-normality data. Therefore, we reanalyzed with the Spearman correlation and corrected our manuscript. The new analysis showed the same significant findings.

Thank you very much again for your helpful comments. 

Best regards, 

Pornjira Pariwatcharakul, MD, MSc, MA, FRCPsychT, MRCPsych (UK)

Associate Professor

Department of Psychiatry

Faculty of Medicine Siriraj Hospital, Mahidol University

2 Wanglang Road, Siriraj, Bangkoknoi, Bangkok 10700, Thailand.

Tel: 0-2419-4293-8, Fax: 0-2419-4298; Mobile: +6681-483-7041

E-mail: pornjira.par@mahidol.edu, pornjirap@gmail.com

---

## [Decision Letter · Decision Letter 1]

25 Feb 2020

Thai psychiatrists and burnout: a national survey

PONE-D-19-31990R1

Dear Dr. Pariwatcharakul,

We are pleased to inform you that your manuscript has been judged scientifically suitable for publication and will be formally accepted for publication once it complies with all outstanding technical requirements.

With kind regards,

Sergio A. Useche, Ph.D.

Academic Editor

PLOS ONE

Additional Editor Comments (optional):

Reviewers' comments:

Reviewer's Responses to Questions

**Comments to the Author**

1. If the authors have adequately addressed your comments raised in a previous round of review and you feel that this manuscript is now acceptable for publication, you may indicate that here to bypass the “Comments to the Author” section, enter your conflict of interest statement in the “Confidential to Editor” section, and submit your "Accept" recommendation.

Reviewer #1: All comments have been addressed

2. Is the manuscript technically sound, and do the data support the conclusions?

Reviewer #1: Yes

3. Has the statistical analysis been performed appropriately and rigorously? 

Reviewer #1: Yes

4. Have the authors made all data underlying the findings in their manuscript fully available?

Reviewer #1: No

5. Is the manuscript presented in an intelligible fashion and written in standard English?

Reviewer #1: Yes

6. Review Comments to the Author

Reviewer #1: All of my comments were addressed. Nonetheless, I believe this manuscript should also be reviewed by a psychiatrist.

7. PLOS authors have the option to publish the peer review history of their article (what does this mean?). If published, this will include your full peer review and any attached files.

Reviewer #1: Yes: Krit Pongpirul

---

## [Editor Report · Acceptance letter]

9 Mar 2020

PONE-D-19-31990R1 

Thai psychiatrists and burnout: a national survey 

Dear Dr. Pariwatcharakul:

I am pleased to inform you that your manuscript has been deemed suitable for publication in PLOS ONE. Congratulations! Your manuscript is now with our production department. 

With kind regards,

on behalf of

Dr. Sergio A. Useche 

Academic Editor

PLOS ONE